# Hypoxia-Regulated CD44 and xCT Expression Contributes to Late Postoperative Epilepsy in Glioblastoma

**DOI:** 10.3390/biomedicines13020372

**Published:** 2025-02-05

**Authors:** Kosuke Kusakabe, Akihiro Inoue, Takanori Ohnishi, Yawara Nakamura, Yoshihiro Ohtsuka, Masahiro Nishikawa, Hajime Yano, Mohammed E. Choudhury, Motoki Murata, Shirabe Matsumoto, Satoshi Suehiro, Daisuke Yamashita, Seiji Shigekawa, Hideaki Watanabe, Takeharu Kunieda

**Affiliations:** 1Department of Neurosurgery, Ehime University School of Medicine, 454 Shitsukawa, Toon 791-0295, Ehime, Japan; kk_0145@yahoo.co.jp (K.K.); y.nkmr6084@gmail.com (Y.N.); y.ohtsuka0818@gmail.com (Y.O.); ma.nishikawa1985@gmail.com (M.N.); shirabem13@gmail.com (S.M.); satoshi.suehiro@gmail.com (S.S.); yamadai551208@gmail.com (D.Y.); shigekawa.seiji@gmail.com (S.S.); whideaki@m.ehime-u.ac.jp (H.W.); takeharukunieda@gmail.com (T.K.); 2Department of Neurosurgery, Washoukai Sadamoto Hospital, 1-6-1 Takehara, Matsuyama 790-0052, Ehime, Japan; takanoriohnishi0024@gmail.com; 3Department of Molecular and Cellular Physiology, Ehime University School of Medicine, 454 Shitsukawa, Toon 791-0295, Ehime, Japan; hjyano@gmail.com (H.Y.); mechoudhury81@gmail.com (M.E.C.); 4Division of Genetic Research, ADRES, Ehime University, 3-5-7 Tarumi, Matsuyama 790-8566, Ehime, Japan; murata.motoki.wv@ehime-u.ac.jp

**Keywords:** brain tumor-related epilepsy, glioblastoma, glutamate, CD44, xCT, hypoxia, glioma stem cells

## Abstract

Background/Objectives: Late epilepsy occurring in the late stage after glioblastoma (GBM) resection is suggested to be caused by increased extracellular glutamate (Glu). To elucidate the mechanism underlying postoperative late epilepsy, the present study aimed to investigate the expressions and relations of molecules related to Glu metabolism in tumor tissues from GBM patients and cultured glioma stem-like cells (GSCs). Methods: Expressions of CD44, xCT and excitatory amino acid transporter (EAAT) 2 and extracellular Glu concentration in GBM patients with and without epilepsy were examined and their relationships were analyzed. For the study using GSCs, expressions and relationships of the same molecules were analyzed and the effects of CD44 knock-down on xCT, EAAT2, and Glu were investigated. In addition, the effects of hypoxia on the expressions of these molecules were investigated. Results: Tumor tissues highly expressed CD44 and xCT in the periphery of GBM with epilepsy, whereas no significant difference in EAAT2 expression was seen between groups with and without epilepsy. Extracellular Glu concentration was higher in patients with epilepsy than those without epilepsy. GSCs displayed reciprocal expressions of CD44 and xCT. Concentrations of extracellular Glu coincided with the degree of xCT expression, and CD44 knock-down elevated xCT expression and extracellular Glu concentrations. Hypoxia of 1% O_2_ elevated expression of CD44, while 5% O_2_ increased xCT and extracellular Glu concentration. Conclusions: Late epilepsy after GBM resection was related to extracellular Glu concentrations that were regulated by reciprocal expression of CD44 and xCT, which were stimulated by differential hypoxia for each molecule.

## 1. Introduction

Glioblastoma multiforme (GBM) is the most malignant brain tumor, characterized by high invasiveness and inter- and intra-cellular heterogeneity of tumor cells. The former allows infiltrating tumor cells to escape from tumor resection and the latter results in the production of tumor cells that are resistant to chemo-radiotherapy. Both features can lead to tumor recurrence and progression, resulting in poor prognosis for GBM patients.

The poor prognosis includes not only short survival but also deterioration of life quality due to common comorbidities such as brain tumor-related epilepsy (BTRE). BTRE is a common symptom in GBM patients and often causes serious clinical deterioration, delays in neurological recovery, and decreased overall survival [1,2,3]. BTRE is recognized not only as a presenting symptom but also as a sign of tumor progression or recurrence [4]. The former is epilepsy at onset, and the latter corresponds to postoperative late epilepsy, which usually occurs several months after tumor resection and often presents as refractory seizure attack, thus resulting in more unfavorable outcomes than GBM patients with epilepsy at onset [5]. However, the causes of the prognostic difference between these types of epilepsy and the mechanisms underlying postoperative late epilepsy remain unsolved. As described above, it is suggested that occurrence of postoperative late epilepsy is related to tumor progression or recurrence in GBM, but no evidence clearly showing such a relationship has been reported. In our previous study, we found that all GBMs presenting postoperative late epilepsy showed a high-invasive type on magnetic resonance imaging (MRI). Also, these GBM highly expressed cluster of differentiation (CD) 44 in the tumor periphery [6].

CD44 is one of the glioma stem cell markers with multifunctional activity, including cell adhesion and signal transductions, resulting in participation in promoting migration and invasion in various cells including glioma stem cells.

We have demonstrated that a multifunctional molecule such as CD44 can not only promote migration and invasion of tumor cells but also participate in the development of tumor recurrence through hypoxia-regulated expression of CD44 [7]. When considering that occurrence of postoperative late epilepsy is a cellular event before development of tumor recurrence in GBM, it will be very important to know how CD44 interacts with potential molecules participating in the seizure attack in postoperative late epilepsy. So, we highlighted the functional role of CD44 as a key molecule in the occurrence of postoperative late epilepsy in GBM. 

Many preclinical and clinical studies have supported a significant role of the excitatory neurotransmitter glutamate (Glu) in BTRE, with Glu deeply participating in the development of epilepsy. Extracellular Glu also enhances the proliferation of malignant brain tumors by binding to the Glu receptor on tumor cells [8]. Accumulated Glu in the extracellular space in peritumoral brain tissue can promote the onset of BTRE [9]. Concentrations of Glu in and around tumor cells are regulated mainly by two Glu transporters. One is the cystine/glutamate antiporter xCT, which releases Glu into the extracellular space by increasing the uptake of cystine into cells in exchange [10]. The other is excitatory amino acid transporter (EAAT) 2, which promotes uptake of Glu into cells [11]. Mediating over 50% of Glu transport in glioma cells, xCT constitutes the main mechanism contributing to high extracellular concentrations of Glu [12,13]. EAAT2 is also a key Glu transporter and is the most abundant subtype of EAAT [14]. Expression of xCT is reportedly up-regulated in GBM [15], and expression of EAAT2 in both tumoral and peri-tumoral tissues has been shown to be downregulated in GBM patients with epilepsy [16].

Considering that epilepsy occurs by excitation of neuronal cells, the peritumoral normal brain tissues may present cues to the onset of BTRE. The area also includes a tumor border niche where GSCs reside under hypoxic conditions. This niche constitutes a tumor microenvironment that includes various factors and participates in regulating the activities of GSCs. Hypoxia is a critical factor that can influence such a tumor microenvironment, including effects of boosting glutamate release from GSCs [17]. In the present study, to clarify differences in the amounts and interactions of extracellular Glu, CD44, xCT, and EAAT2 in the tumor periphery of GBM patients with and without epilepsy, we investigated the expressions of these molecules and analyzed the interrelationships by comparing expressions between GBM tissues collected from the core and periphery in patients with and without epilepsy. In addition, we performed an in vitro study using human GSC lines in which we investigated the relation between postoperative late epilepsy and invasion and proliferation of tumor cells by evaluating the effect of hypoxia on these molecules and extracellular concentrations of Glu to elucidate the roles of these molecules in the development of epilepsy. We reported that alteration of hypoxic conditions in the tumor microenvironment could allow GSCs to induce a critical cellular event of phenotypic transition from migrating cells to proliferating cells [7]. The present studies would present new insights into the molecular mechanisms underlying GBM-related postoperative late epilepsy and could lead not only to identification of novel target molecules for controlling GBM-related epilepsy but also to the development of effective therapeutic strategies against tumor recurrence.

## 2. Materials and Methods

This study was approved by the Ethics Committee for Clinical Research at Ehime University Hospital (approval no. 2110012). All procedures were performed in accordance with the ethical standards of the 1964 Declaration of Helsinki and its later amendments.

### 2.1. Patients and Study Design

Fourteen GBM patients with and without epilepsy, treated at Ehime University Hospital between April 2020 and July 2023, were included in the study. From tumor samples, protein expressions of CD44 and the two Glu transporters, xCT and EAAT2, in the tumor core and periphery were examined. Concentrations of extracellular Glu were also measured from the tumor core and periphery and analyzed. The in vitro study was performed using three human GSC lines, GSC-1, -2, and -3, which had been established by primary culture of tumor tissues obtained from the tumor periphery of three GBM patients as previously described [7]. Expressions of CD44, xCT, and EAAT2 and extracellular Glu concentration were investigated in these GSCs. In addition, the effects of CD44 knockdown on expressions of xCT, EAAT2, and Glu were investigated. Further, to analyze the effects of hypoxia on GSCs, expressions of these molecules were evaluated under severe hypoxia of 1% O_2_ and moderate hypoxia of 5% O_2_.

### 2.2. Imaging Studies

MRI examinations were performed using a 3-T scanner (Achieva; Philips, Best, The Netherlands) with a standard head coil. Axial, coronal, and sagittal T1-weighted images were obtained with a 2 mm slice thickness before and after intravenous administration of gadolinium-diethylenetriamine pentaacetic acid (0.1 nmol/kg). Based on the features observed on MRI, patients were divided into those with high-invasive (HI) and low-invasive (LI) types, as previously described (Appendix A) [18].

### 2.3. Surgical Treatment and Tissue Sampling

All patients underwent craniotomy for tumor resection, followed by radiotherapy (60 Gy) and chemotherapy with temozolomide in accordance with the Stupp protocol [19]. Tumor resection was performed by multimodal navigation-guided microsurgery using fence-post catheter techniques [20]. Tumor samples were separately obtained from the core and periphery of the tumor as previously described [21] (Appendix A).

Samples from the tumor periphery were extracted from the tip of fence-post catheters that were placed along the tumor border using echo-linked image-guided navigation just before starting tumor resection. The target area corresponds to just outside of the Gd-enhanced area and within the positive-uptake area on Methionine-PET. Tumor samples at the core were obtained from the resected tumor mass. All tumor samples were frozen and preserved at −80 °C until use.

### 2.4. Western Blot Analysis

Total proteins were extracted from each tumor sample using RIPA Buffer (188-02453; FUJIFILM Wako, Osaka, Japan) and homogenized by ultrasonication. Extracted proteins were heated at 95 °C for 5 min and loaded on SuperSep^TM^ Ace (195-14951; FUJIFILM Wako) for electrophoresis using a PowerPac Basic power supply (1645050JA; Bio-Rad, Hercules, CA, USA). Proteins were transferred to a polyvinylidene difluoride membrane (033-22453; FUJIFILM Wako) and blocked in 5% skim milk for 10 min before each incubation with the following antibodies: CD44 (#3570; Cell Signaling Technology, Dabverse, MA, USA), xCT (#26864-1-AP; Proteintech, Rosemont, IL, USA), EAAT2 (#22515-1-AP, Proteintech), and beta actin (#66009-1-Ig, Proteintech). All antibodies were used at a dilution of 1:500 except for beta actin, which was used at 1:5000. Images were acquired and quantified by C-digit and Image Studio (LI-COR Biosciences, Lincoln, NE, USA). We made sure that the selection of the area around the band was just around the specific band of interest. Total proteins from GSCs were collected and extracted by cell scraper and RIPA buffer (188-02453; FUJIFILM Wako) after culture to full confluence.

### 2.5. Measurement of Extracellular Glu Concentrations in Tumor Tissues and GSCs

Total amino acid including Glu was extracted from each tumor sample using a 5% solution of 5-sulfosalicylic acid dihydrate (194-04575; FUJIFILM Wako) homogenized by ultrasonication. After centrifugation, the supernatant was collected through a filter (Millex-GV SLGVR33RB; Merck, Darmstadt, Germany). Quantification was carried out by Amino Acid Analyzer (L-8900; Hitachi High-Tech Corporation, Tokyo, Japan).

Total amino acids from GSCs were extracted by collecting the medium previously replaced from the medium described in “GSC culture” to high-glucose Dulbecco’s modified Eagle’s medium (DMEM) without L-glutamine and phenol red (#040-30095; FUJIFILM Wako) supplemented with L-glutamine (073-05391; FUJIFILM Wako) and pyruvic acid (190-14881; FUJIFILM Wako) 24 h before collection. The collected medium was frozen at −80 °C, then freeze-dried by LABCONCO (FZ-2.5; Asahi Life Science, Tokorozawa, Japan). Dried samples were dissolved in methanol then centrifuged, and the collected supernatant was dried with nitrogen gas. Dried samples were dissolved in 20 mM hydrochloric acid, collected through a filter (Millex-GV SLGVR33RB; Merck), and quantified using an Amino Acid Analyzer (L-8900; Hitachi High-Tech Corporation). Results were normalized by cell count.

### 2.6. GSC Culture

The three previously established human GSC lines, GSC-1, -2, and -3 (GSC-1: GSL-1; GSC-2: GSL-3; GSC-3: GSL-2) [6,21], were used in the present study. The stemness of these GSCs was confirmed by sphere-forming activity before performing the experiments. These cell lines were cultured in high-glucose DMEM (043-30085; FUJIFILM Wako) supplemented with penicillin-streptomycin solution (168-23191; FUJIFILM Wako). The contents of the stem cell medium are detailed in the Appendix A. Cells were maintained in an incubator with 5% CO_2_ and 100% humidity at 37 °C. Among these GSCs, GSC-2 showed the highest invasive activity and GSC-1 presented the lowest [6,7,20].

### 2.7. Establishment of Stable CD44-Knockdown Cells

To evaluate the relationship between CD44, extracellular Glu, and the two Glu transporters, we obtained GSC-2*, in which the *CD44* gene in GSC-2 was knocked down with short hairpin (sh) RNA. The details of the procedure are described in the Appendix A.

### 2.8. Inhibition of xCT Expression by the Treatment with Sulfasalazine (SSZ)

To investigate the functional activity of xCT in GSCs, which could protect tumor cells from reactive oxygen species, extracellular Glu and cysteine concentrations were measured and apoptosis status was analyzed in GSCs under xCT inhibition by sulfasalazine (SSZ, S0883-10G, Sigma-Aldrich, Saint Louis, MO, USA). Cysteine, the measurable molecule in this study, is derived from cystine which is transported by xCT. Quantification of Glu and cysteine were conducted as described in “Measurement of extracellular Glu concentrations in tumor tissues and GSCs”. The apoptosis was evaluated by detecting cleaved poly-ADP ribose polymerase (PARP) under moderate hypoxia of 5% O_2_, performed as described in “Western blot analysis”, using anti-polyclonal PARP antibody (#9542; Cell Signaling Technology) in 1:500 dilution. SSZ solution was administered to the GSC culturing dish at concentrations of 200 uM and 400 uM. Ultrapure water was used as a control. The treatments were performed 24 h prior to measurement.

### 2.9. Hypoxic Treatment

To evaluate the dynamics of Glu metabolism and expressions of CD44, xCT, and EAAT2 under different levels of hypoxia, the three GSC lines were maintained under an atmosphere of 1% O_2_, 5% CO_2_, and 94% N_2_ for severe hypoxia and under 5% O_2_, 5% CO_2_, and 90% N_2_ for moderate hypoxia. Cells were incubated for 24 h under each hypoxic condition in a multi-gas incubator (APM-50D; ASTEC, Fukuoka, Japan). Control samples were incubated in the original incubator after medium replacement.

### 2.10. Statistical Analysis

Data were compared using binomial tests. Comparisons among more than two groups were conducted using two-tailed one-way analysis of variance with the Tukey post hoc test. Significance was set at the level of *p* < 0.05. All analyses were performed using Office Excel software (Microsoft 365; Microsoft, Redmond, WA, USA).

## 3. Results

### 3.1. Clinical Features

The 14 patients comprised 4 patients with perioperative epilepsy (Group E) and 10 patients without epilepsy (Group NE). The median age was 74 years (range, 46–85 years) for the 14 patients (66 years for Group E, 75 years for Group NE). Group E included three male patients (75%) and Group NE included four (40%). All tumors were located at the supratentorial region and histopathologically verified as GBM grade IV, isocitrate dehydrogenase wild type, according to the World Health Organization classification. No significant differences between groups were seen for age or sex. Based on MRI findings, nine patients were classified with the high-invasive type of GBM and five with the low-invasive type. All patients but one in Group E presented as high-invasive type. Two patients in Group E developed postoperative epilepsy (Table 1).

### 3.2. Extracellular Glu Concentration in Tumor Tissues of GBM

The extracellular concentrations of Glu were significantly higher in Group E both in the tumor core and in the total tumor tissue compared to Group NE. In the tumor periphery, Glu concentration tended to be larger in Group E than in Group NE (Figure 1).

### 3.3. Expression of CD44, xCT, and EAAT2 in the Tumor Core and Periphery

Expressions of CD44 and xCT in the tumor periphery were significantly higher in Group E than in Group NE (Figure 2). In contrast, expressions of EAAT2 did not differ significantly between Groups E and NE in either the tumor core or periphery (Figure 2).

### 3.4. Extracellular Glu Concentration and Expressions of CD44, xCT and EAAT2 in GSCs

Among the three GSC lines, GSC-2, demonstrating the highest migration and invasion ability in the previous studies [6,21], presented the highest expression of CD44 and the lowest expression of xCT. In addition, GSC-2 showed the lowest extracellular Glu concentration (Figure 3). On the other hand, GSC-1, with the lowest invasiveness, presented the lowest expression of CD44 and higher expression of xCT than GSC-2. GSC-1 showed higher extracellular Glu than GSC-2. GSC-3 showed relatively high expression of xCT and the highest amount of extracellular Glu. EAAT2 expression was low in all GSCs but the expression pattern of EAAT2 was similar to that of CD44 and inverse to that of xCT (Figure 3). As a result, extracellular Glu concentrations in these GSCs correlated with the degree of xCT expression and correlated inversely with expressions of CD44 and EAAT2.

### 3.5. Effects of CD44 Knockdown on Expressions of xCT and EAAT2, and Extracellular Glu Concentrations

To investigate the role of CD44 on the two Glu transporters and extracellular Glu, knockdown of the *CD44* gene with shRNA was performed on GSC-2, which presented the highest expression of CD44. Knockdown of CD44 in GSC-2 significantly increased the expressions of xCT and extracellular concentrations of Glu. No significant differences in EAAT2 were identified (Figure 4).

### 3.6. Inhibitory Effects of Sulfasalazine (SSZ) on xCT

Inhibition of xCT by SSZ decreased extracellular Glu release and increased extracellular cysteine concentration in all GSCs (Figure 5a). In addition, treatment of GSCs with SSZ promoted the cleaved PARP under 5% O_2_ hypoxic conditions, demonstrating the induction of apoptosis by SSZ in all GSCs (Figure 5b) (Appendix A).

### 3.7. Effects of Differential Hypoxia on Expressions of CD44, xCT, EAAT2, and Glu Concentration in GSCs

Extracellular Glu concentration tended to decrease under severe hypoxia at 1% O_2_ in all GSCs. In contrast, extracellular Glu concentration tended to increase under moderate hypoxia compared to severe hypoxia in all GSCs, especially GSC-2, showing a significant increase (Figure 6a). On the other hand, a tendency was seen for severe hypoxia to increase CD44 and EAAT2 expressions and decrease xCT expression in all GSCs compared to moderate hypoxia. The changes in CD44 and xCT expression between the two conditions of hypoxia were significant only in GSC-2 (Figure 6b) (Appendix A).

## 4. Discussion

BTRE is a major unfavorable event in patients with GBM and can occur at any stage during the course of the disease. In particular, postoperative late epilepsy differs from the preoperative epilepsy at onset in terms of both underlying mechanisms and patient prognosis. Postoperative late epilepsy has been suggested to relate to tumor progression and recurrence based on the fact that it generally occurs with a certain interval (>21 days) after tumor resection [5], but the definite relation between postoperative late epilepsy and tumor recurrence has not been clarified. In the present study, to elucidate the mechanisms underlying postoperative epileptogenesis in GBM, we focused on the expressions and functions of CD44, xCT, and EAAT2 as potential molecules closely related to late epilepsy. CD44 is known to have multiple functions, including promotion of tumor migration, invasion, proliferation, and recurrence. On the other hand, xCT can primarily regulate the extracellular release of Glu and is known to be upregulated in GBM [22,23]. Extracellular accumulation of Glu can promote not only excitotoxicity in neuronal cells but also tumor growth through the binding of Glu to its receptor in an autocrine fashion [24]. These molecules expressed on GSCs may regulate not only tumorigenesis at GBM recurrence but also postoperative epileptogenesis in GBM.

In the present study, peritumoral Glu concentration in GBM was significantly higher in Group E than in Group NE. Similarly, expressions of CD44 and xCT in the peritumoral area were higher in Group E than in Group NE. In contrast, expressions of EAAT2 did not differ significantly between the two groups. These results indicate that expressions of CD44 and xCT were upregulated in the peritumoral area of GBM patients with the occurrence of epilepsy. In addition, most patients with epilepsy displayed high-invasive type GBM on MRI.

To elucidate the functional roles of these molecules in GBM epileptogenesis, we performed an in vitro study using human GSC lines. The GSC lines showed different patterns of expression for CD44, xCT, and EAAT2. The results demonstrated that extracellular concentrations of Glu coincided with the expression pattern of xCT, whereas CD44 and EAAT2 displayed an inverse pattern. Following this, to examine the relationship among these three molecules, we obtained GSC-2* by knock down of the *CD44* gene on GSC-2 with shRNA, which presented the highest expression of CD44. GSC-2* showed significantly increased expressions of xCT along with elevated extracellular Glu compared to GSC-2. These results suggest that CD44 and xCT may be reciprocally expressed. As EAAT2 functions to restore Glu within cells, increased levels may participate in lowering extracellular concentrations of Glu. However, considering the small amount of EAAT2 expression by GSCs, effects on the extracellular concentration of Glu may not have been significant compared to those of xCT.

The theoretical dichotomy expressed by “migrating cells cannot proliferate” [25] may be applied to the present relationship between CD44, as a stimulator for cell migration, and xCT, as a promotor for cell proliferation. Most recurrence of GBM occurs locally around the tumor resection cavity. GSCs, which reside in the tumor border niche, migrate into the peritumoral area, and, surviving resection and radio-chemotherapy, may wait for the opportunity to cause tumor recurrence [26]. After the injured tissues of the cavity wall have been repaired, the highly migratory GSCs may return to the original site around the resection cavity guided by the gradient of oxygen tension, where the environment is of moderate hypoxia of 2.5–5%O_2_ [27]. In such a microenvironment of moderate hypoxia, GSCs can aggressively proliferate because this particular condition of hypoxia provides a favorable oxygen condition for proliferation of cancer stem cells [28]. Here, we display an illustration of hypothetical mechanisms that shows common cellular processes shared in developing tumor recurrence and postoperative late epilepsy in GBM under a gradient of differential hypoxia (Figure 7).

Considering the time duration required for repairment at the resection site, not only tumor recurrence but also postoperative late epilepsy is unlikely to occur in the earlier stage after surgery. In another group of our 23 GBM patients, seven patients presented postoperative late epilepsy. The median time from surgery to occurrence of epilepsy was 75 days (ranging 34 to 240 days) and that from surgery to recurrence was 282 days (ranging from 42 to 1257 days). All patients with postoperative late epilepsy presented epilepsy before tumor recurrence (Appendix A). In addition, patients who presented a shorter time both between tumor resection and late epilepsy onset and between epilepsy onset and detection of tumor recurrence tended to show shorter survival than patients who displayed longer intervals between these events (Appendix A).

As described above, moderate hypoxia provides favorable oxygen conditions for cancer stem cell proliferation, and it also reduces CD44 expression, as observed in the present study, decreasing the highly migratory and invasive activities of GSCs. On the other hand, the proliferation of GSCs activated by moderate hypoxia promotes excessive production of reactive oxygen species (ROS). To protect GSCs from such oxidative cytotoxicity, intracellular uptake of cystine increases though activating xCT, leading to the synthesis of glutathione as an antioxidant. As a result, the activated xCT releases large amounts of Glu in exchange for cystine uptake. These functional activities of xCT were demonstrated in our present study, in which GSCs were treated with an xCT inhibitor, sulfasalazine (SSZ). These cellular processes could also be explained from the perspective of energy metabolism. We investigated the cellular energy metabolism of the three GSC lines by evaluating adenosine triphosphate (ATP) production and mitochondrial respiration (Seahorse XF HS Mini Analyzer; Agilent Technologies, Santa Clara, California, USA) under hypoxic conditions of 1% and 5% O_2_ and normoxia (Appendix A). All GSCs showed increased ATP production and basal respiration at 5% O_2_ compared to 1%, suggesting increased mitochondrial functions by this hypoxic alteration.

As described earlier, activities of GSCs on tumorigenesis and epileptogenesis could be influenced by the oxygenation of the microenvironment. We therefore investigated expressions of CD44, xCT, and EAAT2 under two different hypoxic conditions, including severe hypoxia with 1% O_2_ and moderate hypoxia with 5% O_2_. CD44 expression was upregulated under 1% O_2_ in all GSCs but tended to decrease under 5% O_2_. In contrast, xCT expression was decreased by 1% O_2_ and increased by 5% O_2_ in all GSC lines. EAAT2 expression tended to decrease under 5% O_2_. In all GSCs, extracellular Glu concentrations decreased with hypoxia under 1% O_2_ and elevated under 5% O_2_. These results indicate that severe hypoxia at 1% O_2_ increases CD44, resulting in promotion of higher migration and invasion of GSCs, whereas moderate hypoxia at 5% O_2_ increases xCT expression, resulting in accumulation of extracellular Glu, leading to promotion of postoperative epileptogenicity in GBM as well as tumor proliferation (Figure 6). Although inhibition of the *CD44* gene resulted in increased expression of xCT, precisely how these molecules participate in cellular functions in GSCs or how they interact with each other remains unclear. CD44 variant isoforms reportedly bind to xCT on the cellular membrane and stabilize xCT expression in mouse gastric cancer stem cells, increasing the uptake of cystine as an essential substrate for glutathione synthesis, preventing toxicity by ROS [29]. However, in GSCs, whether CD44 variant isoforms have the same activity as observed in mouse tumor cells or what variant isoforms of CD44 participate in such functions of xCT are not known. Although these processes may become new targets in the treatment of GBM, the present study observed no such enhancing effects of CD44 on the activity of xCT. These results indicate that CD44 may have other roles in functional activity through interactions with xCT, or the shRNA used in the present study for silencing CD44 gene may not inhibit the expression of specific CD44 variant isoforms that stimulate the activity of xCT.

EAAT2 is also a major Glu transporter [14]; however, the present study suggested that its effect on extracellular Glu concentration was minimal. Furthermore, CD44 knockdown in GSCs did not have a significant effect on EAAT2, while hypoxia under 1–5% O_2_ decreased EAAT2 expression in all GSCs. Hypoxia of 1–5% O_2_ could affect EAAT2 towards elevating extracellular Glu concentrations, but the effects may be smaller than those of xCT.

The present study had clear limitations in not demonstrating the actual appearance of epileptic attack in an animal model using transplanted GSCs with knock-down xCT or inhibition of epileptic attack in a model with xCT overexpression by administration of an xCT inhibitor, such as SSZ [30]. Additional studies are required to elucidate the relationship between CD44 and the Glu transporters, xCT, and EAAT2 and the association with CD44 isoforms to further understand the mechanisms underlying postoperative late epilepsy. This study indicated that epileptogenicity in malignant glioma requires the co-existence and reciprocal expression of CD44 and xCT, so both could be potential targets for anti-epileptic and anti-tumoral therapies in GBM. However, because this was a single-center study conducted by analyzing data from a relatively small cohort of patients, it was not possible to explore potential confounders in clinical data.

The present study potentially provided new targets to control late epilepsy after resection of GBM. An α-amino-3-hydroxy-5-methyl-4-isoxazolepropionic acid (AMPA) receptor antagonist is now seeing increasing clinical use as an anti-epileptic drug, also often being used against BTRE. In addition, various reports have described the antitumoral effects of AMPA receptor antagonist [31,32,33,34]. The mechanisms of this effect are explained in varied ways, such as a reduction in cell metabolism due to decreased glucose uptake [31] and induction of apoptosis and synergistic effects with temozolomide [32], but the full definitive mechanism remains unclear. In the future, further elucidation of the mechanisms underlying postoperative late epilepsy could provide new therapeutic approaches for treating not only epilepsy but also tumor recurrence in GBM.

## 5. Conclusions

In the present study, tumor tissues obtained from GBM patients of Group E showed higher expression of CD44 and xCT in the tumor periphery than Group NE. Group E also showed higher levels of extracellular Glu than Group NE. In an in vitro study using human GSC lines, expressions of CD44 and xCT were reciprocal and expression patterns of xCT coincided with those of extracellular Glu. Knockdown of *CD44* in GSC-2, which expressed the highest level of CD44, significantly elevated xCT expression and extracellular Glu concentration. Considering the postoperative alteration of oxygen condition at the resection site, we investigated effects of differential hypoxia on expressions of these molecules. Hypoxic alteration of 1 to 5% O_2_ decreased CD44 expression while increasing xCT in all GSC lines. EAAT2 expression decreased and extracellular Glu increased with this alteration of hypoxia in all GSCs. These results indicate that GBM-related postoperative epilepsy is accompanied by increased accumulation of extracellular Glu which resulted from the reciprocal reactions of CD44 and xCT. Differential hypoxia may regulate this cellular process.

## Figures and Tables

**Figure 1 biomedicines-13-00372-f001:**
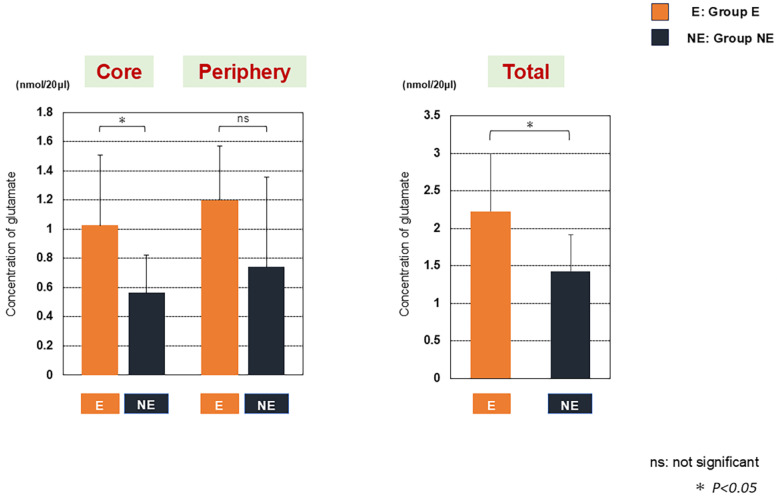
Concentrations of glutamate (Glu) in the tumor core and periphery of glioblastoma (GBM) samples with perioperative epilepsy (Group E) and without epilepsy (Group NE). Extracellular concentration of Glu was significantly larger in Group E, both in the tumor core and for total tumor tissue, than Group NE. In the tumor periphery, the Glu concentration in Group E tended to be larger compared to Group NE. ns, not significant; * *p* < 0.05; E, Group E; NE, Group NE.

**Figure 2 biomedicines-13-00372-f002:**
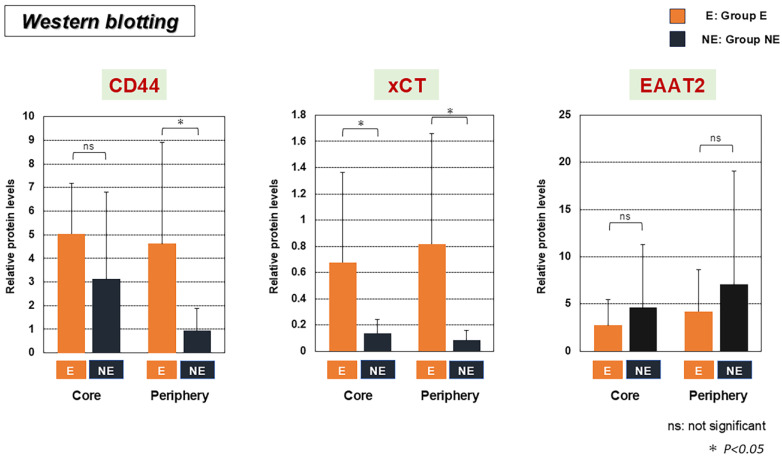
Western blotting analysis comparing expressions of cluster of differentiation 44 (CD44), xCT, and excitatory amino acid transporter 2 (EAAT2) between Group E and NE in the tumor core and periphery. Group E showed higher expression of CD44 in the periphery and higher xCT in both the core and periphery of the tumor. No significant difference was seen for EAAT2. ns, not significant; * *p* < 0.05; E, Group E; NE, Group NE.

**Figure 3 biomedicines-13-00372-f003:**
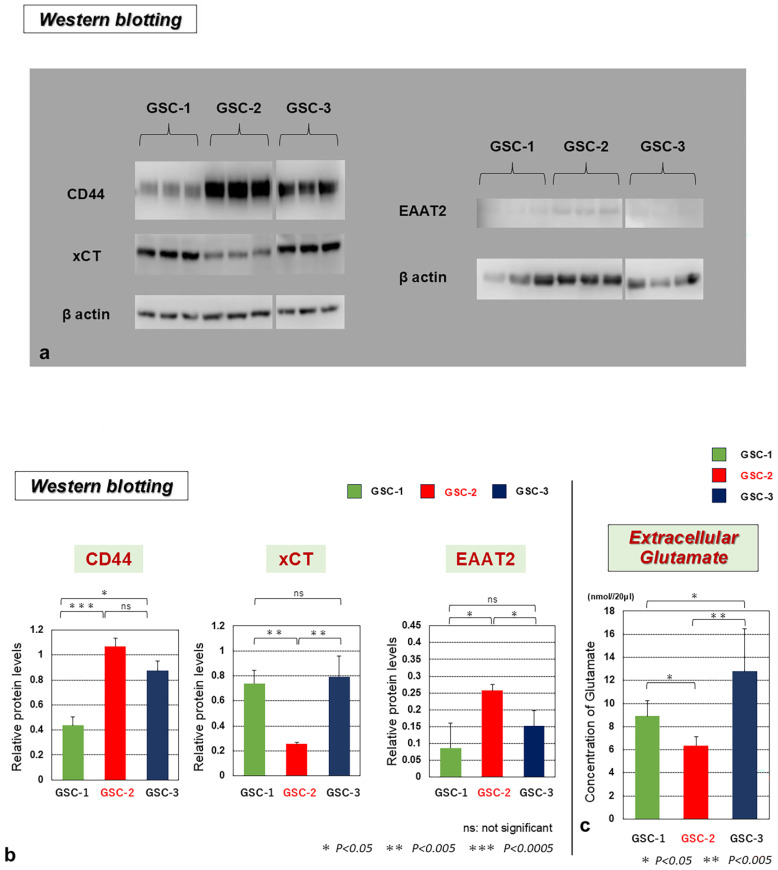
(**a**) Western blotting analysis of CD44, xCT, and EAAT2 expression showing on the gel. (**b**) Expressions of CD44, xCT, and EAAT2 and (**c**) extracellular Glu concentration in three glioma stem-like cell (GSC) lines, GSC-1, -2, and -3. GSC-2 showed the highest CD44 expression while displaying the lowest xCT and extracellular Glu concentration and the highest EAAT2. The pattern of extracellular Glu concentration coincided with that of xCT. Protein levels were normalized as ratios versus β-actin and indicated as “Relative protein levels”. ns, not significant; * *p* < 0.05; ** *p* < 0.005; *** *p* < 0.0005.

**Figure 4 biomedicines-13-00372-f004:**
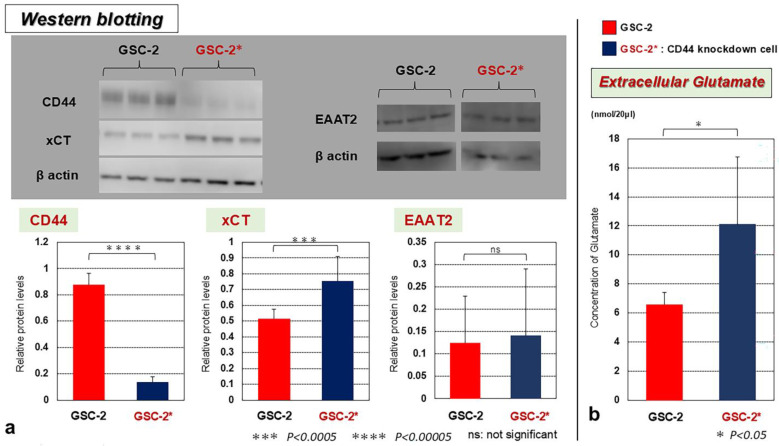
(**a**) Effects of *CD44* knockdown on expressions of xCT and EAAT2 and (**b**) extracellular Glu concentration in GSC-2. GSC-2* (stable *CD44*-knockdown cells) displayed an increased expression of xCT and also elevated extracellular Glu. No significant differences in EAAT2 were seen with CD44 knockdown. Protein levels were normalized as ratios versus β-actin and indicated as “Relative protein levels”. ns, not significant; * *p* < 0.05; *** *p* < 0.0005; **** *p* < 0.00005.

**Figure 5 biomedicines-13-00372-f005:**
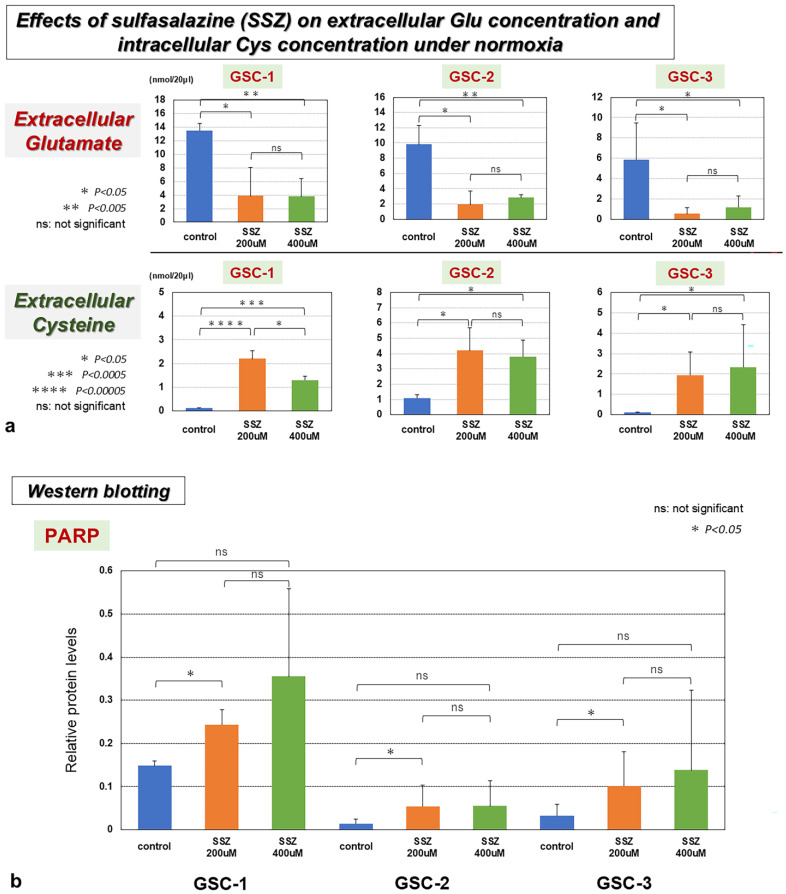
(**a**) Inhibition of xCT by sulfasalazine (SSZ) decreased extracellular Glu release and increased extracellular cysteine concentration in all GSCs. (**b**) Treatment of GSCs with SSZ promoted the cleaved poly-ADP ribose polymerase (PARP) under 5% O_2_ hypoxic conditions, demonstrating the induction of apoptosis in all GSCs. ns, not significant; * *p* < 0.05; ** *p* < 0.005; *** *p* < 0.0005; **** *p* < 0.00005.

**Figure 6 biomedicines-13-00372-f006:**
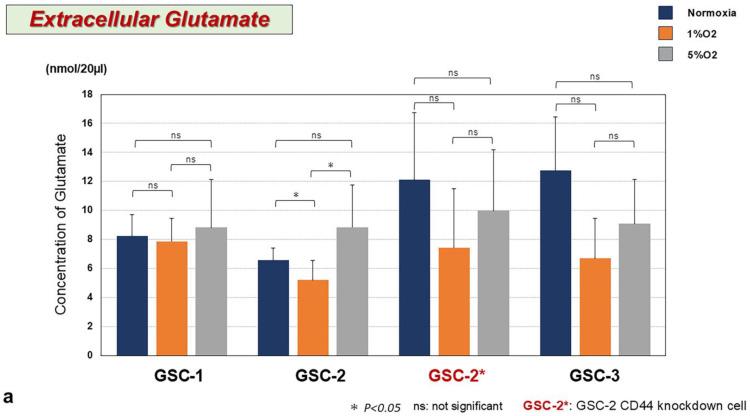
(**a**) Extracellular Glu concentrations in the three GSC lines and GSC-2* under differential hypoxia of 1% and 5% O_2_. All GSCs showed a tendency toward an increase in Glu concentration by changing the hypoxia from 1% to 5% O_2_. In particular, GSC-2 was the only cell line to demonstrate a significant increase in Glu. (**b**) Effects of hypoxia (1% and 5% O_2_) on expressions of CD44, xCT, and EAAT2 in the three GSC lines. A tendency toward severe hypoxia increasing CD44 and EAAT2 expression and decreasing xCT expression was seen in all GSC lines compared to moderate hypoxia. Changes of CD44 and xCT expression between the two hypoxias were significant only in GSC-2. ns, not significant; * *p* < 0.05, *** *p* < 0.0005.

**Figure 7 biomedicines-13-00372-f007:**
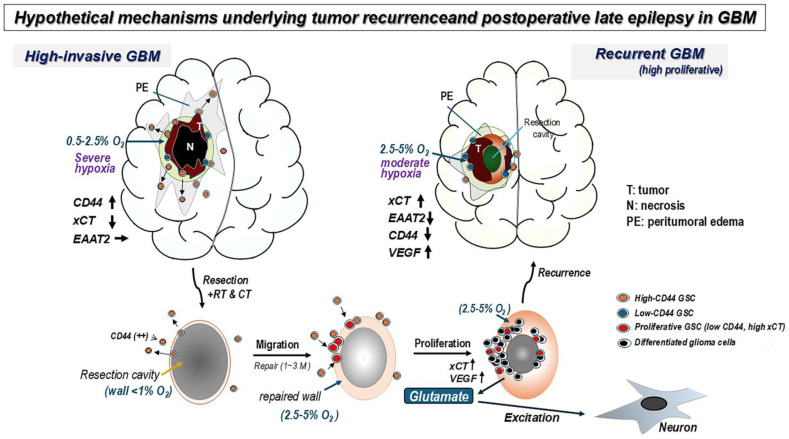
Hypothetical mechanisms underlying tumor recurrence and postoperative late epilepsy in GBM, with the two processes developing coordinately. After resection of high-invasive GBM, infiltrating GSCs, abundant with CD44, in the peritumoral area tend to remain and survive. These GSCs migrate toward the original tumor site around the resection cavity as tissue repairment progresses accompanied by altering hypoxia of 1% to 5% O_2_, prompted by activating vascular endothelial growth factor (VEGF). This area of moderate hypoxia provides a favorable condition for proliferation of cancer stem cells. Aggressive proliferation of GSCs leads to accumulation of reactive oxygen species (ROS). Consequently, xCT up-regulates to reduce the cytotoxicity while expression of CD44 decreases, demonstrating the converting phenotypes from high-invasive and high-migratory to low-invasive and high-proliferative tumor cells. Along with this, activated xCT releases excessive Glu in the extracellular space, which is expected to promote epileptogenicity in the pre-stage of tumor recurrence. To achieve this cellular behavior, co-existence and reciprocal expressions of CD44 and xCT should be required.

**Table 1 biomedicines-13-00372-t001:** Summary of GBM samples including data of Glu, xCT, EAAT2, and CD44. Group E is highlighted in orange.

Sample	Age (y)	Sex	Epilepsy (Pre-Op)	Epilepsy (Post-Op)	Image Type	Glu			xCT		EAAT2		CD44	
						Core	Peri	Total	Core	Peri	Core	Peri	Core	Peri
1	71	M	(+)		H	0.863	1.406	2.269	1.696	2.013	3.875	8.888	6.49	1.69
2	70	F	(+)		L	1.649	1.616	3.265	0.289	0.284	2.206	7.048	6.986	0.548
3	74	M			L	0.415	0.311	0.726	0.297	0.107	9.77	5.569	1.103	1.259
4	49	M			H	0.321	1.831	2.069	0.044	0.024	3.957	7.481	8.757	0.297
5	73	F			L	0.638	0.868	0.882	0.047	0.047	1.824	2.168	6.815	2.674
6	62	M	(+)	(+)	H	1.088	0.849	1.938	0.254	0.176	1.259	0.711	2.32	6.483
7	76	F			L	1.045	0.089	1.089	0.131	0.032	1.482	1.363	1.449	0.493
8	74	F			L	0.429	0.497	0.926	0.224	0.228	1.147	1.706	0.606	0.831
9	85	F			H	0.531	0.834	1.365	0.052	0.042	1.229	0.285	3.11	0.171
10	75	F			H	0.535	0.895	1.495	0.084	0.057	0.911	0.285	0.012	0.003
11	81	M			H	0.271	0.424	0.696	0.37	0.069	0.082	1.605	7.467	3.021
12	46	M	(+)	(+)	H	0.501	0.93	1.431	0.467	0.794	0.77	0.185	4.398	9.76
13	60	F			H	0.418	2.812	3.23	0.044	0.019	0.006	0.143	0.617	0.232
14	76	M			H	1.011	0.8945	1.905	0.002	0.004	0.242	0.18	0.645	0.456

GBM, glioblastoma multiforme; Glu, glutamate; EAAT2, excitatory amino acid transporter 2; CD44, cluster of differentiation 44; M, male; F, female; pre-op, preoperative; post-op, postoperative; Peri, periphery; H, high-invasive type; L, low-invasive type.

## Data Availability

All data used for analysis are presented in the tables in this article.

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
