# Peer review of "Hypoxia-Regulated CD44 and xCT Expression Contributes to Late Postoperative Epilepsy in Glioblastoma"

_biomedicines, 2025, doi:10.3390/biomedicines13020372_

Round 1

Reviewer 1 Report

Comments and Suggestions for Authors

The manuscript explores the molecular mechanisms underlying postoperative late epilepsy in glioblastoma, highlighting the reciprocal roles of CD44 and xCT under hypoxic conditions, but methodological flaws and insufficient analyses must be addressed to reach publication standards. Specifically:

1.      The introduction lacks a clear articulation of the research gap and hypothesis, requiring refinement to explicitly highlight the study's novelty and the gap it addresses in understanding postoperative epilepsy in glioblastoma.

2.      The study's inclusion of only 14 GBM patients limits its statistical power and generalizability, necessitating justification through power calculations or expansion of the cohort.

3.      Some statistically significant results lack biological or clinical relevance, such as minor EAAT2 expression changes, requiring a focus on clinically meaningful findings and their contextual significance.

4.      The oversimplified relationship between oxygen levels (1% vs. 5%) and molecular changes needs mechanistic insights, supported by complementary experiments like mitochondrial assays to clarify the observed effects.

Reviewer 2 Report

Comments and Suggestions for Authors

The article titled “Postoperative late epilepsy in glioblastoma occurs by hypoxia-promoted reciprocal expression of CD44 and xCT in infiltrating glioma stem cells” studied the extracellular Glu concentrations on late epilepsy after GBM resection, this article is well written and informative, but some concerns need to be addressed:

1.      From table 1, there are only 2 post-op epilepsy samples, and the authors just used these two experimental samples in their research, the conclusion is hard to believe due to the small sample size, can the authors explain?

2.      The detail information about how the core and peri area was divided should be described in the methods section.

3.      The authors should introduce if there are any literature that have studied the relationship between Glu and hypoxia.

4.      a. Does xCT express in CD44 positive cells?

b.     In figure 2, expressions of CD44 in the tumor core has no significant difference, this is different from the xCT changes. If xCT expressed in CD44 positive cells, these results were not consistent. The authors should give some discussion about this result.

5.      In Figure 3a and Figure 4 right panel, are these cropped WB images from the same membrane? If yes, please provide the original images. If not, the internal reference proteins run on different WB membranes cannot be compared together, therefore, no conclusion can be drawn from this result.

Reviewer 3 Report

Comments and Suggestions for Authors

This paper presents an important and well-structured study to investigate the molecular mechanisms of late postoperative epilepsy in glioblastoma (GBM). The focus on the reciprocal expression of CD44 and xCT under hypoxic conditions is both novel and clinically relevant. The study is commendable for its comprehensive approach, combining clinical data and in vitro experiments to help understand tumor biology and potential therapeutic strategies.

However, several aspects need to be improved, and the following are specific suggestions:

1. Please indicate the sample size and the method of significant difference analysis in the figure caption. Use appropriate labels for figures. Use a, b, c for each graph.

2. The title could be more concise. For example: "Hypoxia-regulated CD44 and xCT expression contributes to late postoperative epilepsy in glioblastoma."

3. In Figure 1 part, please specify whether the differences in extracellular Glu concentration between Groups E and NE were clinically significant.

4. In Figure 4 part, please add a brief explanation of why EAAT2 did not show significant changes. Provide more detail on why EAAT2 expression remains unchanged despite its known role in glutamate transport.

5. In discussion, please include a dedicated section on limitations and future directions, e.g., validation in animal models, exploration of CD44 isoforms, or potential confounders in clinical data.

6. Provide brief guide for supplementary materials in the main text.

Round 2

Reviewer 1 Report

Comments and Suggestions for Authors

All the concerns raised have been duly resolved by the authors and the respevtive changes have been implemented. Now, the manuscript is eligible for publication in its current form. 

Author Response

We thank the reviewer for the kind comments.

We hope you will give this manuscript favorable consideration and find it worthy of publication in your journal. Please feel free to contact me if you have any questions or require further information.

Reviewer 2 Report

Comments and Suggestions for Authors

Although the authors addressed most of my comments, the WB results in this article are highly academically inaccurate. “Relative protein levels,” in channels on different WB membranes cannot be compared. Such results are very unreliable, the authors should provide new WB evidence to support their conclusion.

Round 3

Reviewer 2 Report

Comments and Suggestions for Authors

The authors addressed all my comments. This manuscript is ready for publication.